# Immunohistochemical Identification and Assessment of the Location of Immunoproteasome Subunits LMP2 and LMP7 in Acquired Cholesteatoma

**DOI:** 10.3390/ijms241814137

**Published:** 2023-09-15

**Authors:** Justyna Rutkowska, Irena Kasacka, Marek Rogowski, Ewa Olszewska

**Affiliations:** 1Department of Otolaryngology, Medical University of Bialystok, 15-089 Białystok, Poland; marek.rogowski@umb.edu.pl (M.R.); ewa.olszewska@umb.edu.pl (E.O.); 2Department of Histology and Cytophysiology, Medical University of Bialystok, 15-089 Białystok, Poland; kasacka@umb.edu.pl

**Keywords:** LMP2, LMP7, immunoproteasome, cholesteatoma, immunohistochemistry

## Abstract

Cholesteatoma, accompanied by chronic inflammatory response, is characterized by invasive growth and osteolytic activity. As specific proteasome isoforms, the immunoproteasomes serve as an important modulator of inflammatory responses. The aim of the present study was to determine the biological activity of cholesteatoma through the analysis of the expression and localization of immunoproteasome subunits of low molecule weight protein (LMP) 2 and LMP7. Cholesteatoma specimens were obtained from 15 adults who underwent ear surgery due to acquired attic cholesteatoma. Normal skin specimens were taken from retro-auricular skin incisions from the same patients. The specimens were stained with anti-LMP7 antibody, using immunohistochemistry techniques based on the binding of biotinylated secondary antibody with the enzyme-labeled streptavidin and the Envision FLEX system. In all specimens of cholesteatoma, the immunohistochemical reaction with the antibody against the LMP2 was positive, in both the cytoplasm of the cholesteatoma matrix and the perimatrix. A negative reaction with anti-LMP2 was observed in the cytoplasm and nuclei of control skin cells. A positive nuclear and cytoplasmic immunohistochemical reaction with anti-LMP7 has been demonstrated in numerous cells, in both the matrix and perimatrix of cholesteatoma. We present evidence of the presence of expressions of LMP2 and LMP7 within cholesteatoma tissue. Our results might bring new information concerning immunoproteasome-dependent pathophysiologic mechanisms in cholesteatoma.

## 1. Introduction

Middle ear cholesteatoma (MEC) is a non-neoplastic cystic formation composed of the hyperproliferative stratified keratinizing squamous epithelium (matrix), subepithelial connective tissue (perimatrix) and an accumulation of desquamated keratin and squamous debris in the middle ear or mastoid region. MEC accompanied by a chronic inflammatory response is characterized by invasive growth and osteolytic activity with local bone destruction. It is important to distinguish acquired cholesteatoma from congenital cholesteatoma, which is present at birth and arises from trapped embryonic epithelial cells. Acquired cholesteatoma is a condition that results from a combination of factors affecting the middle ear, with still-unknown triggers for epithelial hyperplasia, disturbed cell migration, inflammatory responses in the preexisting retraction pocket or tympanic membrane perforation. Acquired cholesteatoma is divided into primary, arising in tympanic membrane retraction, and secondary, with the epithelial migration through the perforated tympanic membrane.

The EAONO/JOS (European Academy of Otology and Neurotology/Japanese Otological Society) staging system applies to pars flaccida cholesteatoma, pars tensa cholesteatoma, congenital cholesteatoma, and cholesteatoma secondary to a tensa perforation: Stage I: Cholesteatoma localized in the primary site (the site of cholesteatoma origin, i.e., the attic (for pars flaccida cholesteatoma); the tympanic cavity (for pars tensa cholesteatoma).Stage II: Cholesteatoma involving two or more sites.Stage III: Cholesteatoma with extracranial complications or pathologic conditions including facial palsy, labyrinthine fistula with conditions at risk of labyrinthitis, postauricular abscesses or fistula, zygomatic abscesses, and neck abscesses [1].

Despite the extensive research that has been carried out so far, a significant number of issues remain controversial. The networks of molecular interactions underlying cholesteatoma pathogenesis, including high proliferative activity, inflammatory response, cell signaling disruptions, neovascularization, apoptosis, the cell cycle, cell differentiation, bone resorption, and the remodeling process, remain still unclear. The treatment of choice is surgery, but the role of new potential molecular targets of therapeutic handle are constantly being studied.

The regulation of protein homeostasis in a cell, including numerous pro-apoptotic and anti-apoptotic protein degradation processes, occurs within a multi-subunit complex known as proteasome [2]. The proteasomal degradation pathway may affect the proteolytic cleavage of the extracellular bone matrix adjacent to cholesteatoma. Three catalytically active β subunits (β1, β2, and β5) in proteasomes stimulated by interferon-γ (IFN-γ) or tumor necrosis factor-α (TNF-α) are replaced by the inducible catalytic subunits LMP2 (β1i), MECL-1 (β2i), and LMP7 (β5i), building the immunoproteasomes. The immunoproteasome is a form of the constitutive proteasome that is expressed in nonimmune cell types during inflammation or neoplastic transformation, supporting a role in the pathogenesis of inflammatory, autoimmune diseases, and neoplastic diseases [3,4]. Immunoproteasomes were identified as factors in the pathomechanism underlying the inflammatory process [5,6]. Recent clinical investigations have indicated that immunoproteasomes are necessary factors for T-helper cell differentiation and the control of pathogenic immune responses [5,6].

They are involved in the response to protein-damaging conditions, participating in the removal of oxidized proteins [7]. Although immunoproteasomes are critical for antigen presentation, they also appear to be a link between inflammatory factors and the control of apoptosis [8]. The enhanced activity of the immunoproteasomes prevents the accumulation of degradation substrates that would otherwise aggregate during inflammation [9,10]. Study data suggest that immunoproteasome subunits may regulate the degradation of some pro-apoptotic proteins and influence the cell death molecular process [2].

According to the current state of knowledge, inflammation, oxidative stress, and cellular imbalance associated with impaired cellular proliferation and apoptosis are considered to be critical for the development and progression of cholesteatoma [11,12,13]. Therefore, it seems to be crucial to evaluate the significance of the immunoproteasome subunits LMP2 and LMP7 in cholesteatoma. We hypothesize that the expression of immunoproteasome subunits LMP2 and LMP7 may be associated with the accumulation of degradation substrates that aggregate during inflammation responses in cholesteatoma. The aim of the study was to pinpoint the presence and localization of immunoproteasome subunits LMP2 and LMP7 within the cholesteatoma tissue. The study’s findings may contribute to understanding the immune responses and molecular processes associated with cholesteatoma development and progression and potentially identify targets for therapeutic interventions or diagnostic markers. To our knowledge, this is the first report on the expression of immunoproteasome subunits LMP2 and LMP7 in cholesteatoma.

## 2. Results

There were 10 female and 5 male patients with a mean age of 50 years. Clinical basal characteristics are presented in Table 1. All patients complained of periodic or constant otorrhea and deterioration of hearing. Four patients complained of preoperative vertigo and dizziness. Facial paralysis was recognized as a complication of cholesteatoma in one case. Labyrinthitis due to a lateral semicircular canal fistula caused by cholesteamatous otitis media was diagnosed in three patients. The cholesteatoma was limited to the attic region in four patients (26.67%); attic with antrum in seven (46.67%); and attic with antrum and middle ear in four patients (26.67%). Moreover, ossicular chain erosion was observed in all cases (100%). Incus and maleus head erosion were the most common. Partially absorbed ossicles with an adherent cholesteatoma and granulation tissue were completely dissected out during surgical procedure. Intraoperatively, seven ears (46.67%) demonstrated an exposed facial nerve. Exposed dura mater was present in six (40%) patients. Slight granulation tissue was observed behind the cholesteatoma in 7 of the 15 patients (46.67%). Amongst the surgically treated patients, ten (66.67%) underwent attico-antro-mastoidectomy, two (13.34%) attico-antrotomy, and one (6.67%) modified radical mastoidectomy, and radical mastoidectomy was performed in two (13.34%) patients. 

### 2.1. LMP2 Expression

There was no significant difference in the expression of LMP2 among specimens from patients belonging to different subgroups based on age, sex, and clinical stage. In all specimens (*n* = 15) of cholesteatoma, the immunohistochemical reaction with the antibody exhibiting LMP2 was positive, in the cytoplasm of both the cholesteatoma matrix and the perimatrix (Figure 1c). The most intensive immunohistochemical reaction and strong staining in grayscale were detected in the cytoplasm of perimatrix cells (Table 2). The strongest reaction in the cholesteatoma matrix with anti-LMP2 was observed in the basal layer. There was no immunohistochemical reaction in the nuclei of the cholesteatoma matrix and perimatrix cells (Table 2). As is shown in Figure 1b,d , an immunohistochemical reaction with anti-LMP2 was negative in the nuclei and cytoplasm of cells of the normal epidermis and dermis. An immunohistochemical reaction in vascular endothelial cells in cholesteatoma was observed (161.9 ± 3.10 in grayscale), in contrast to the skin, where cells were LMP2 negative.

### 2.2. LMP7 Expression

There was no significant difference in the expression of LMP7 among specimens from patients belonging to different subgroups based on age, sex, and clinical stage.

A positive nuclear and cytoplasmic immunohistochemical reaction with anti-LMP7 has been demonstrated in numerous cells, in both the matrix and perimatrix of the cholesteatoma (Figure 2a,c). Based on the assignment of the intensity of staining, the strongest reaction was observed in the nuclei of the cholesteatoma matrix and the cytoplasm of cholesteatoma perimatrix cells (Table 2). There was weak cytosolic staining in the cholesteatoma matrix, but greater compared to a healthy epidermis (Table 2). The immunohistochemical reaction was not observed in the nuclei in the skin epidermis (Figure 2b, Table 1). In specimens of the dermis, single LMP7 positive cells were presented (weak staining) (Figure 2d). Staining in vascular endothelial cells in the dermis was less intense when compared to perimatrix cholesteatoma (194.8 ± 2.99 versus 141.9 ± 5.35 in grayscale).

## 3. Materials and Methods

### 3.1. Study Samples

Fifteen ears in fifteen adult patients (aged 27–74; mean age = 50) with chronic otitis media with acquired attic cholesteatoma were operated on using a microscopic approach. Cholesteatoma (a study group, *n* = 15) and macroscopically healthy retro-auricular skin specimens (a control group, *n* = 15) were obtained from the same adult patients during middle ear surgery. The disease was classified according to the EAONO/JOS staging system. Based on the staging system, 4 patients were included in Stage I, 7 patients in Stage II, and 4 patients in Stage III. Middle ear cholesteatoma, as well as specimens of retro-auricular skin, were obtained during the same surgery. Specimens were immediately fixed in 10% buffered formalin and routinely embedded in paraffin. Sections (4 µm) were stained with hematoxylin-eosin for histological examination and processed with immunohistochemistry to detect local LMP2 and LMP7 immunoproteasome subunits’ immunoreactivity.

All patients included in the study were treated in the Clinic for Otorhinolaryngology, Bialystok, Poland. The study protocol was approved by the Ethics Committee (approval number: R-I-002/366/2014). Written informed consent had previously been obtained from each subject. All patients signed the agreement to participate in the study.

### 3.2. Immunohistochemistry

Immunohistochemical reactions were performed on 4 µm paraffin sections. The EnVision method was used, according to Herman GE and Elfont EA [14]. Immunostaining was performed using the following protocol: paraffin-embedded sections were deparaffined and hydrated in pure alcohols. For antigen retrieval, the sections were subjected to pretreatment in a pressure chamber and heated for 1 min at 21 psi at 125 °C, using Target Retrieval Solution with pH of 9.0 (S 1699) for LMP2 antibody and Target Retrieval Solution Citrate at pH 6.0 (S2369 Dako Denmark A/S, Produktionsvej 42, DK-2600 Glostrup) for LMP7. After cooling the sections down to room temperature, they were incubated with Peroxidase Blocking Reagent (S 2001 Dako Denmark A/S, Produktionsvej 42, DK-2600 Glostrup) for 10 min to block endogenous peroxidase activity. The sections with the primary antibody (Proteasome 20S β1i subunit mouse monoclonal antibody (LMP2–13) purchased from Enzo Life Sciences cat. number BML-PW8840) and 20S proteasome subunit β 5i (LMP7) (PW 8845 Affinity Research Products, Enzo Biochem Executive Offices, Farmingdale, NY, USA) were incubated for 24 h at +4 °C in a humidified chamber. The antisera were previously diluted in Antibody Diluent (S 0809 Dako Denmark A/S, Produktionsvej 42, DK-2600 Glostrup) to 1:1000 (LMP2) and 1:5000 (LMP7). The procedure was followed by incubation (1 h) with secondary antibody (conjugated to horseradish peroxidase-labeled polymer) (EnVision+ Kit HRP Mouse K4007 Dako Denmark A/S, Produktionsvej 42, DK-2600 Glostrup). The bound antibodies were visualized using 1 min incubation with liquid 3,3′-diaminobenzidine substrate chromogen. The sections were finally counterstained in hematoxylin QS (H-3404, Vector Laboratories; Burlingame, CA, USA), mounted and evaluated under a light microscope. Washing with Wash Buffer (S 3006 Dako Denmark A/S, Produktionsvej 42, DK-2600 Glostrup) was performed between each step (3 times for 5 min). Control reactions were simultaneously performed. Sections from the same tissue served as negative controls, i.e., the primary antibody was omitted. Sections from lymph nodes were used as positive controls.

### 3.3. Quantitative Analysis

The intensity of all immunohistochemical reactions was estimated independently via blinded examiner (I.K, pathologist) using an Olympus BX43 light microscope (Olympus 114 Corp., Tokyo, Japan) with an Olympus DP12 digital camera (Olympus 114 Corp., Tokyo, Japan) and documented. Five randomly selected microscopic fields (each field had 0.785 mm^2^, 200× magnification (20× lens and 10× eyepiece)) from each section were documented using an Olympus DP12 microscope camera. Cells with the immunoexpression of analyzed antigens LMP2 and LMP7 as a result of reaction with antibodies, anti-LMP2 and anti-LMP7 (intensely nuclear or cytoplasmic brown staining), were determined as antigen positive. Cells with no immunohistochemical reaction were called antigen negative. The intensity of immunohistochemical reaction was measured by using 0 to 256 gray scale level, where a completely black pixel received a value of 0, whereas one with a value of 256 was completely white or bright. Based on the assignment of the intensity of staining using the scoring scale from 256 to 0, the following categories of the IHC reaction results were distinguished: strong staining (0–149 in grayscale), moderate strong staining (150–179 in grayscale), weak staining (180–229 in grayscale), and negative staining (230–256 in grayscale).

## 4. Discussion

In this study, the expressions of LMP7 and LMP2 immunoproteasome subunits were found to be upregulated in the acquired cholesteatoma by comparing with those of the retro-auricular skin of the patients. It is noteworthy that for the first time, we showed the upregulations of these molecular factors. The multistep pathway of cholesteatoma development connected with LMP2 and LMP7 involves events in the nucleus and cytoplasm. The expression of LMP7 and LMP2 immunoproteasome subunits is related, among other things, to an inflammatory process, apoptosis, and oxidative stress in tissues, but these factors have not yet been explored in cholesteatoma [4,8,15]. 

The literature data indicate that immunoproteasomes formation is dependent on stimulation by tumor necrosis factor-α (TNF-α) or interferon-γ (IFN-γ) [8,15]. Due to the important role of inflammatory molecules, such as TNF-α, that are highly expressed in cholesteatoma and closely related to the progression of adultacquired cholesteatoma, it can be assumed that immunoproteasomes’ activity also plays a role in otitis media [16,17,18]. The TNF-α is localized in all layers of the cholesteatoma matrix and is considered to have a vital role in bone resorption [17,19]. A strong correlation was detected between TNF-α and bone resorption in congenital and acquired cholesteatoma [18].

Recent studies prove that the NF-κB signaling pathway controls the expression of genes including inflammatory cytokines, chemokines, and adhesion molecules. They play pivotal roles in controlling inflammation and may be linked with the role of the LMP7 immunoproteasome subunit [20,21]. In the inflammatory phase of cholesteatoma, nuclear factor-kappa B (NF-κB) causes the overexpression of microRNA-802, affecting keratinocyte cell proliferation and cell cycle progression [22]. In normal skin, only basal keratinocytes can proliferate and differentiate into mature cells. They undergo nucleation to generate the cornified layer and NF-κB is found in the cytoplasm of basal keratinocytes only [23]. In cholesteatoma, NF-κB expression is observed mostly in the nucleus, localized in the basal layer, and also in the suprabasal layers [23]. Byun et al. suggested that there might be an imbalance between the anti-apoptotic role and the cell cycle inhibitory role of NF-κB, whereby the scale is tipped toward protection against cell death in the context of a constitutive cytokine-rich inflammatory milieu. The activated NF-κB may protect against apoptosis and allow for the hyperproliferation seen in cholesteatoma [23]. It was also found that the LMP7 is crucial for the suppression of caspase-independent cell death, and with the normal activation of the proapoptotic transcription factor p53 is protective against caspase-independent cell death [7]. LMP7 and LMP2 positive cells were observed in all layers of the cholesteatoma matrix, which may indicate their role in the impaired proliferation, differentiation, and cell death in keratinocytes. It may be possible that keratinocyte migration within layers of the cholesteatoma matrix, as well as inflammation within its perimatrix, is related to the enhanced expression of LMP7, especially in basal layers. To better understand the relationship between inflammation and the expression of immunoproteasomes, additional, more extensive studies should be conducted. Correlation studies to analyze the levels of immunoproteasome subunits in relation to various inflammatory markers (cytokines, chemokines) can help to establish the strength and consistency of the relationship.

It is worth emphasizing that the LMP2 not only initiates, amplifies, and perpetuates inflammation, but is also is involved in apoptosis [7,8]. In the literature, reports show the important role of apoptosis in cholesteatoma development [24]. It was noticed that a dysfunction of apoptosis may contribute to the accumulation of oxidized proteins during aging and in age-related diseases [25]. The inhibition of apoptosis in acute promyelocytic leukemia cells has led to an increase in the level of LMP2 immunoproteasome [25]. It was observed that a variety of solid cancer cell lines, as well as prostate tumor tissues, express high levels of LMP2 [4]. A depletion of the LMP2 subunit in PC-3 (prostate cancer cell line) has a growth inhibitory effect, supporting LMP2 as a novel target in conditions with an impaired process of apoptosis [4]. Moreover, it was shown that deleting LMP2 in murine embryonic fibroblasts greatly reduced BIM (the pro-apoptotic protein family member that regulates the final cell death checkpoint in developing thymocytes and peripheral T- and B-cells) degradation [2]. It was demonstrated that the LMP2-containing proteasome was required for the degradation of proteins involved in the initiation of apoptosis, and consequently that LMP2-containing proteasomes protected cells from undergoing lymphocytes apoptosis [2]. In our study, LMP2 was expressed only within cholesteatoma tissue, not in normal retro-auricular skin. This suggests that LMP2 may serve as a potential marker for cholesteatoma. However, it is important to validate LMP2 as a marker rigorously. If these validation studies confirm that LMP2 is a specific and reliable marker for cholesteatoma, it could have implications for improving the diagnosis and management of the condition.

The regulatory mechanism of LMP2 on the expression and activities of MMP-2 and MMP-9 has been proved [26,27]. It was observed that MMP-9 and MMP-2 overexpression may be implicated in the molecular inflammatory pathways in cholesteatoma development [12,28,29]. LMP2 contributes to IkappaBalpha degradation and p50 generation, and the inhibition of LMP2 suppresses the expression and activities of MMP-2 and MMP-9 by blocking the transfer of active NF-kappaB heterodimers into the nucleus [27]. The mRNA expression of MMP-2 and MMP-9 and their activities were markedly decreased in the LMP2-inhibited cells [27]. In our study, the upregulation of immunoproteasomes with LMP2 and LMP7 in cholesteatoma may be connected with the maintenance and propagation of the inflammatory process in this ear pathology.

Experimental data have revealed that the immunoproteasomes also appear to be a link between inflammatory factors and the control of vascular cell apoptosis [8]. We have noticed an expression of LMP2 and LMP7 subunits in the vascular structures in cholesteatoma. Yang et al. suggest a mechanism for the apoptotic sensitivity associated with immunoproteasome activity induction [8]. They suspect that the role of the immunoproteasome in apoptosis could be related to the degradation of apoptotic inhibitors such as Mcl-1. A siRNA knockdown of LMP7 and overexpression of Mcl-1 and pro-apoptotic effects of IFNγ via the immunoproteasome were observed. LMP2’s and LMP7’s influence on apoptosis in cholesteatoma is possible and it requires further research.

Immunoproteasomes may also be significant for oxidative-stress-related processes. The expression of LMP7 was shown to be crucial for resistance against oxidative stress [9]. Oxidative stress plays a role in the pathogenesis of cholesteatoma [12]. The damage to stromal cells and the proteasomes’ release from the cytoplasm of cells in cholesteatoma may indicate protective protein aggregation occurring during inflammation to maintain protein homeostasis [9]. The possible participation of the immunoproteasome with the LMP7 subunit in oxidative-stress-related processes in cholesteatoma may be suspected.

## 5. Conclusions

The present study has taken the first steps to highlight the potential role of immunoproteasomes in acquired attic cholesteatoma in adults using the immunohistochemical method. We present evidence of the presence of the expression of LMP2 and LMP7 in cholesteatoma. LMP2 may be explored as a diagnostic marker for cholesteatoma. Cholesteatoma is immunologically active tissue. Possible connections between the expression of LMP2 and LMP7 in cholesteatoma with processes such as angiogenesis, cell proliferation, keratinocyte differentiation, apoptosis, and oxidative stress have been mentioned, but require further research. We propose that future studies adopt RNA-based molecular techniques and analysis of the correlation between immunoproteasomes and inflammatory biomarkers to provide insights into the mechanisms underlying immune responses and inflammation in cholesteatoma. New evidence could lead to a better understanding of the pathogenesis of cholesteatoma. The focus of future research work in this area may also concentrate on establishing the selective inhibitors of immunoproteasome subunits LMP2 and LMP7 to eliminate the progression of the disease and recurrent or residual cholesteatoma cases. 

## Figures and Tables

**Figure 1 ijms-24-14137-f001:**
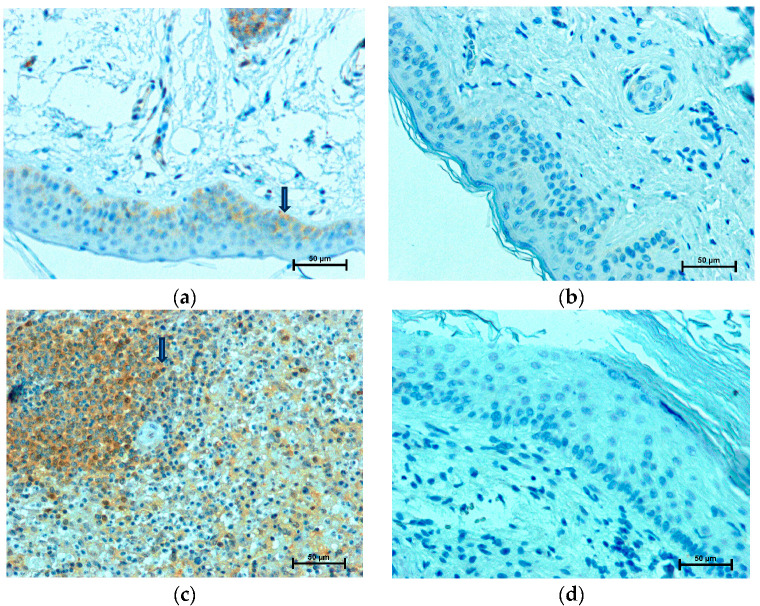
LMP2 expression; LMP2 immunohistochemistry positive cells (arrowhead): (**a**) in cholesteatoma matrix, (**b**) in epidermis, (**c**) in cholesteatoma perimatrix, and (**d**) in dermis.

**Figure 2 ijms-24-14137-f002:**
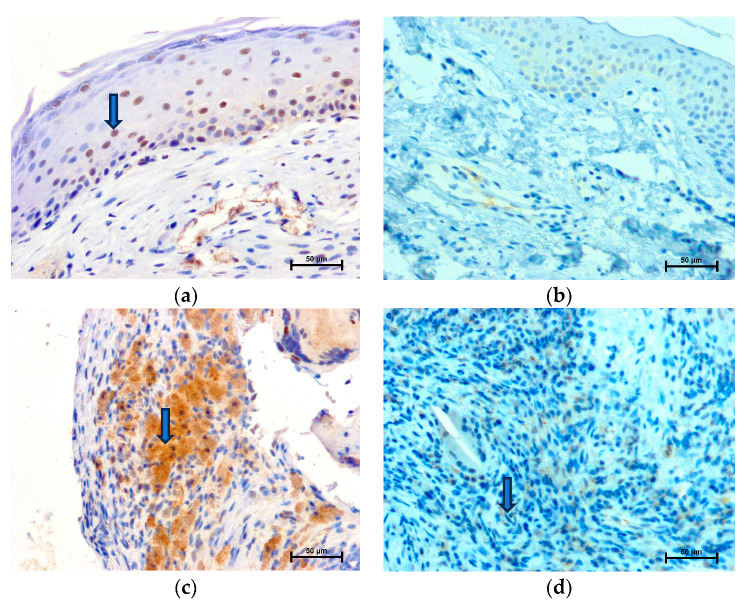
LMP7 expression; LMP7 immunohistochemistry positive cells (arrowhead): (**a**) in cholesteatoma matrix, (**b**) in epidermis, (**c**) in cholesteatoma perimatrix, and (**d**) in dermis.

**Table 1 ijms-24-14137-t001:** Patients’ clinical characteristics.

Gender	Age	Ossicular Chain Destruction	The EAONO/JOS Staging System
female	31	absent of maleus and incus with intact stapes	III (Labyrhintitis)
female	28	erosion of maleus and incus with intact stapes	I
female	41	erosion of maleus and incus with intact stapes	I
female	59	erosion of maleus and incus with intact stapes	I
male	57	erosion of maleus and incus with erosion of stapes	I
female	69	erosion of maleus and incus with intact stapes	III (Facial palsy)
male	57	erosion of maleus and incus with intact stapes	III (Labyrinthine fistula)
female	56	absent of incus with erosion of maleus and intact stapes	II
male	27	erosion of maleus and incus with intact stapes	II
female	28	erosion of maleus and incus with intact stapes	II
female	57	absent of incus with erosion of maleus and erosion of stapes	III (Labyrinthine fistula)
male	74	absent of maleus and incus with erosion of stapes	II
female	66	absent of maleus, incus and stapes	II
female	56	erosion of maleus, incus and stapes	II
male	44	erosion of maleus and incus with intact stapes	II

**Table 2 ijms-24-14137-t002:** LMP2 and LMP7 expression (immunoreactivity in grayscale).

	LMP2 Cytoplasm	LMP2 Nuclei	LMP7 Cytoplasm	LMP7 Nuclei
Cholesteatoma Matrix	161.9 ± 3.10	No reaction	210.9 ± 2.39	125.7 ± 3.21
Control Skin Epidermis	No Reaction	No reaction	222.4 ± 1.55	No reaction
Cholesteatoma Perimatrix	141.3 ± 2.42	No reaction	131.0 ± 2.71	No reaction
Control Skin Dermis	No Reaction	No reaction	181.6 ± 2.60	No reaction

## Data Availability

Data available from the corresponding author on request.

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
