# Peer review of "Immunohistochemical Identification and Assessment of the Location of Immunoproteasome Subunits LMP2 and LMP7 in Acquired Cholesteatoma"

_ijms, 2023, doi:10.3390/ijms241814137_

Round 1

Reviewer 1 Report

Authors reported that the expression of immunoproteosome subunits, LMP2 and LMP7 in cholesteatoma by using immunohistochemistry technique. They also showed the clinical characteristics of each patient. However, this is a new study but requires a few additional experiments to further prove the role of immunoproteasome subunits in cholesteatoma. The following major concerns would further improve the quality of manuscript.

Comments:

1. Authors would utilize these cholesteatoma samples for extraction of RNA and protein to show the levels of immunoproteasome subunits, LMP2 and LMP7.

2. It would be important to show the expression levels of LMP2 and LMP7 in cytoplasm and nuclear fractions of cholesteatoma samples using immunoblotting.

3. Cholesteatoma is accompanied with chronic inflammatory response. Therefore, it would be essential to study the expression of inflammatory markers and its correlation with the immunoproteasome subunits.

4. It is not clear that why authors aim to detect only the expression of immunoproteasome subunits in cholesteatoma samples.

Minor english editing is required

Author Response

For research article: " Immunohistochemical identification and assessment of the location of immunoproteasome subunits LMP2 and LMP7 in acquired cholesteatoma "

Response to Reviewer Comments

Dear Reviewer,

Thank you very much for taking the time to review this manuscript. Please find the detailed responses below and the corresponding revised manuscript/corrections highlighted/.I have carefully considered your comments and suggestions, and I believe they will significantly contribute to improving the quality and clarity of the work. I have made the necessary revisions based on your feedback, and I'm eager to share the updated version with you for your further assessment.

Comment 1:  Authors would utilize these cholesteatoma samples for extraction of RNA and protein to show the levels of immunoproteasome subunits, LMP2 and LMP7.

Response 1: Thank you for pointing this out. We agree with this comment. It would have been interesting and important to explore this aspect. Extracting RNA from cells is a fundamental step in molecular biology and genetic research. However, this research will encompass longer time to reach next steps. We will include this idea as our direction for future research. In the revised manuscript this response can be found: page number 10, line 375-379.

Comment 2:  It would be important to show the expression levels of LMP2 and LMP7 in cytoplasm and nuclear fractions of cholesteatoma samples using immunoblotting.

Response 2: Thank you for this suggestion. EnVision system, is a rapid, highly sensitive two-step immunohistochemical technique and does not lead to false-positive reaction due to the endogenous biotin. It was used as a first step of detection and visualization specific antigens in cholesteatoma sections. Western blot will help to confirm the presence and quantity of proteins through hybridization with specific antibodies. Immunoblotting will be used for analysis of proteins extracted from tissues and will provide information about protein expression, size, and modifications. Our future work will include immunoblotting in a research plan.

Comments 3:  Cholesteatoma is accompanied with chronic inflammatory response. Therefore, it would be essential to study the expression of inflammatory markers and its correlation with the immunoproteasome subunits.

Response 3: We completely agree with this suggestion. You have raised an important point here. Following stimulation with IFN-γ, LPS (lipopolysaccharide) subunits X, Y, and Z in proteasome are replaced by “immunoproteasome” subunits LMP7, LMP2, and MECL-1, respectively. Several pro and anti-inflammatory mediators may be the research objectives. It's important to note that these biomarkers can provide valuable information, but they are not specific to cholesteatoma and may be elevated in various inflammatory conditions. Measurement of inflammatory markers is essential, but broad variety of inflammatory cytokines requires a planning new comprehensive research project. We plan to improve this and develop new further research utilizing correlations between immunoproteasome subunits and inflammatory markers. We mention about this idea in the revised manuscript:  page number 3, line 63; page number 8. Line: 314-318; page number 10, line 375-379.

Comments 4:  It is not clear that why authors aim to detect only the expression of immunoproteasome subunits in cholesteatoma samples.

Response 4: This paper presents an initial study undertaken as the first step in research concerning the immunoproteasome subunits in cholesteatoma. To our knowledge, this is the first report on the expression of immunoproteasome subunits LMP2 and LMP7 in cholesteatoma. The main goal was to pinpoint the presence and spatial information about immunoproteasome subunits LMP2 and LMP7 distribution within cells. This study is designed to investigate a very specific research question to maintain the study's focus and clarity. It is a preliminary research, we start by measuring a single point of interest to determine if there is a promising direction for further investigation. Once initial results are obtained, more comprehensive studies can be designed. We consider the limitations of a single-point measurement and we acknowledge the potential for missing important aspects of a complex phenomenon. Subsequent studies may explore additional points or variables to provide a more understanding of the topic.

In response to comment, we have addressed language issues in the manuscript. English editing was performed to enhance the clarity and readability of the text.

Sincerely,

Justyna Rutkowska

Reviewer 2 Report

Review of the manuscript entitled: Immunohistochemical identifcation and assessment of the location of immunoproteasome subunits LMP2 and LMP7 in acquired cholesteatoma.

In abstract and introduction clear aim of the manuscript should be added e.g. "The aim of the present study was to ...". In case of introduction aim should be at the end of introduction. In the abstract I found this sentence, but I don't see it in the introduction.

Line 94-95 - we do not describe the results in the introduction.

The methodology and results are well described.

Lines 339-340 require references.

In fig the magnification and scale bar should be added.

The entire manuscript should be formatted. I find different font sizes and the text is out of alignment in places.

Author Response

Response to Reviewer Comments

Dear Reviewer,

Thank you very much for taking the time to review this manuscript. Please find the detailed responses below and the corresponding revised manuscript/corrections highlighted/.I have carefully considered your comments and suggestions, and I believe they will significantly contribute to improving the quality and clarity of the work. I have made the necessary revisions based on your feedback, and I'm eager to share the updated version with you for your further assessment.

-In abstract and introduction clear aim of the manuscript should be added e.g. "The aim of the present study was to ...". In case of introduction aim should be at the end of introduction.  In the abstract I found this sentence, but I don't see it in the introduction.

RESPONSE: We have included this sentence according your comment. In abstract and introduction clear aim of the manuscript was added.

- Line 94-95 - we do not describe the results in the introduction:

RESPONSE:  we have removed this text fragment In the revised manuscript this response can be found: page number, line 365-367.

- The methodology and results are well described. RESPONSE: Thank you.

- Lines 339-340 require references.

RESPONSE: Reference 27 is for lines 339-340.

- In fig the magnification and scale bar should be added.

RESPONSE: It was improved.

- The entire manuscript should be formatted. I find different font sizes and the text is out of alignment in places.”

RESPONSE: The entire manuscript  was formatted.

Yours Sincerely

Justyna Rutkowska

Round 2

Reviewer 1 Report

I have gone through the revised version of the manuscript and satisfied with their responses. Therefore, i recommend this manuscript for publication in present form.